# Evaluation of scientific and technological innovation efficiency at Shandong Vocational and Technical University of international studies: A two-stage DEA model approach

Jun Zeng[1]*, Tao Sun[1], Li Zhang[2]

**1** Shandong Vocational and Technical University of International Studies, Rizhao, Shandong, China,
**2** Qingdao Vocational and Technical College of Hotel Management, Qingdao, Shandong, China

* zj916sy@163.com

## Abstract

Vocational undergraduate education is an important part of the higher education system, and science and technology innovation is a major function of vocational undergraduate education. Based on the theory of innovation value chain, the scientific and technological innovation activities of colleges and universities can be divided into two stages: knowledge innovation and results transformation. In this paper, we built the shared-input-type correlation DEA (Data envelopment analysis) model of additional inputs, measured the basic data of Shandong Vocational and Technical University of International Studies during the period of 2013–2022 after its schooling went into a formalized way, and estimated the trend information of the scientific and technological research efficiency of Shandong Vocational and Technical University of International Studies based on the comprehensive consideration of the intermediate output reinvestment, the distribution of initial inputs in the sub-stage and the new inputs in the second stage. Our key findings are: the scientific research input-output efficiency of Shandong Vocational and Technical University of International Studies shows a rapid growth trend, from 0.6194 in 2013 to 0.9091 in 2022, and its scientific research efficiency growth rate has increased by 46.77%, with an average annual growth rate of 4.68%. This increase in research efficiency is crucial, as it enhances the institution's ability to contribute to regional economic development, improves research productivity, and strengthens its role in advancing vocational education. On this basis, we propose targeted policy recommendations, including increasing funding allocation for R&D activities, improving resource utilization efficiency, introducing high-level research talents, and enhancing the alignment between research outputs and industry needs through curriculum adjustments and stronger university-industry collaboration.

**Data availability statement:** All relevant data are within the paper and its Supporting Information files.

**Funding:** The author(s) received no specific funding for this work.

**Competing interests:** The authors declare that there are no conflicts of interest regarding the publication of this paper. The authors have no financial, consultative, institutional, or other relationships that might lead to a conflict of interest.

## 1. Introduction

Scientific and technological innovation is an important strategic support to promote rapid economic and social development and improve the comprehensive strength of the country. Since the 18th CPC National Congress, China has adhered to the strategy of innovation-driven development, improved the capacity for independent innovation, and boosted the construction of a modernized economic space.This commitment reflects the nation's ambition to achieve high-quality development and global competitiveness.

The journey towards this goal began to take shape in May 2019, when the Ministry of Education announced the first 15 pilot vocational undergraduate institutions in the country [1]. Vocational undergraduate education, as an integral part of the modern vocational education system, requires a solid foundation in knowledge and theory. Through scientific and technological innovation, it can establish its unique position within undergraduate-level higher education, which is a critical prerequisite for its high-quality development.

Building on this momentum, in September 2020, General Secretary Xi Jinping highlighted at a symposium of scientists that the world is undergoing profound and complex changes not seen in a century. He emphasized that, in the context of a new round of scientific and technological revolution and industrial transformation, accelerating the pace of innovation is essential to building a strong socialist modernization country. To achieve this, he underscored the importance of harnessing the unique role of colleges and universities in advancing scientific research [2].

Further emphasizing this trajectory, in December 2020, the Ministry of Education convened a national college and university scientific and technological work conference. At this event, it was reiterated that colleges and universities play an irreplaceable role in achieving scientific and technological self-reliance. With the nation entering a new stage of development, the conference stressed the need to comprehensively promote high-quality development in university-led scientific and technological innovation [3].

Scientific and technological innovation activities of colleges and universities is a systematic value chain process, which includes the single direct path from "study and research" to "production", as well as the feedback path from "production" to "study and research". The feedback path of "learning and research" and the circular effect. Based on this, the scientific and technological innovation activities of universities are divided into two stages, namely, knowledge innovation and achievement transformation, which reflect the ability of universities to produce scientific and technological achievements such as theses and patents, and to realize the academic or economic value of the knowledge achievements, respectively.

The efficiency of scientific and technological innovation in colleges and universities refers to the proportional relationship between inputs and outputs of scientific and technological innovation activities of colleges and universities over a period of time (Detailed information is shown in Fig 1.), reflecting the ability of colleges and universities to transform scientific research inputs into scientific research outputs and rationally allocate scientific research resources [4]. Although the overall

## Two-Stage STI Value Chain Process in Universities

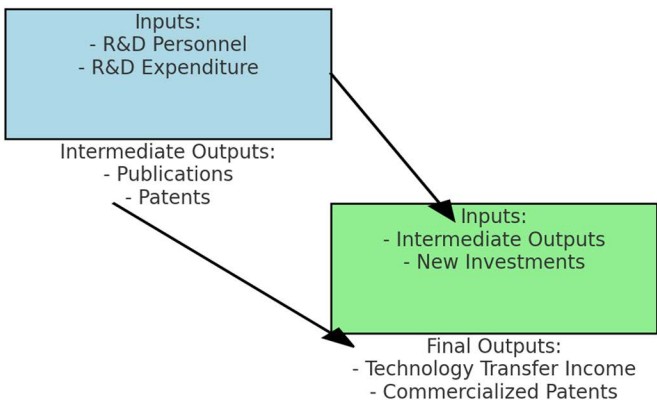

**Fig 1. Two-stage STI value chain process in Universities.**

development level of scientific and technological innovation in colleges and universities in China has been significantly improved, this progress can be observed through key indicators such as the increase in R&D funding, the number of published academic papers, and patent applications over the past decade. According to statistics from the Ministry of Education, the total expenditure on R&D in Chinese universities increased from RMB 300 billion in 2013 to over RMB 520 billion in 2022, representing an average annual growth rate of approximately 6.2%. Furthermore, the number of SCI-indexed papers published by Chinese universities rose from 250,000 in 2013–420,000 in 2022, while patent applications more than doubled during the same period, reaching 150,000 in 2022 [5]. There are still problems such as low utilization of innovation resources and unbalanced regional development [6].

This figure illustrates the two-stage value chain process of scientific and technological innovation (STI) in universities. In the first stage, inputs such as R&D personnel and R&D expenditure produce intermediate outputs, including publications and patents. These intermediate outputs, combined with new investments, serve as inputs for the second stage, which generates final outputs such as technology transfer income and commercialized patents. This model reflects the efficiency of transforming research inputs into outputs and highlights the importance of rational allocation of research resources in universities.

In order to improve the scientific and technological innovation ability of vocational undergraduate colleges, it is necessary to evaluate their scientific and technological innovation efficiency scientifically and objectively, analyze the development of scientific research activities in different regional colleges and universities, so as to clarify the main reasons for the inefficiency of innovation, to provide a direction for the colleges and universities to effectively utilize scientific research inputs, optimize the allocation of resources, maximize the positive effects of innovation, improve the level of scientific and technological innovation outputs, and to provide an empirical evidence for the formulation of governmental relevant policies.

## 2. Research on the efficiency of scientific and technological innovation in colleges and universities

Scientific and technological innovation is a complex process, which is jointly influenced by a variety of factors, such as the economic environment, institutional environment, etc. Abramo et al., [7] empirically analyzed the impact of high-level

researchers on the scientific research performance of Italian universities, and found that the stronger the research and development ability of R&D personnel, the greater the impact on the scientific research efficiency of universities. Bornmann et al., [8] used a multilayer regression model to compare the research efficiency of universities in different countries, and found that the differences in the level of efficiency mainly come from the different levels of economic development.

Benedetto et al., [9] studied the determinants of research performance in Italian universities, and found that both the quality of researchers and the amount of research funding have an important impact on the quality of research. Kolympiris and Klein [10] investigated the impact of academic incubators on innovation activities in universities and came to conclusion that the establishment of university-affiliated incubators may reduce the quality of innovation.

Yeo [11] verifies the impact of university characteristics and regional industrial R&D activities on technological innovation and commercialization in universities by studying the impact of university characteristics and regional industrial R&D activities on technological innovation and commercialization, and verifies the impact of university technology transfer personnel, technology transfer offices, cooperation with high-technology firms, and government financial support on the outputs of university innovations and the technology commercialization.

Domestic scholars mostly adopt the DEA-Tobit (Data envelopment analysis-Tobit) model to study the influencing factors of technological innovation efficiency in universities. For example, Fan and Yu [12] selected the promotion of universities themselves, the support of local governments, the relationship between governments and universities, the absorptive capacity of local enterprises, and the relationship between enterprises and universities as the main factor and the main relationship factor, selected regional GDP as the environmental factor, and used the input and expenditure of relevant funds as the measurement index to study its impact on the efficiency of technology transfer in universities. The results showed that universities' R&D expenditure and enterprise capital investment have a significant promotion effect on technology transfer.

Qin et al., [13] selected regional endowment, degree of opening up to the outside world, industrial structure, degree of financial development, international cooperation, quality of R&D talents and the distribution status of scientific research projects as the influencing factors. The obtained results of the study showed that in addition to the degree of financial development on the efficiency of scientific research achievements of colleges and universities to inhibit the role of the other factors have a positive effect, and only the regional endowment and the quality of R&D talents through the significance test.

Wang et al., [14] selected human capital, physical capital, degree of external connection, R&D institutions and so on as the influencing factors, and found that except for the degree of external openness, all other factors have a significant effect on the scientific research and innovation efficiency of universities. Liu and Xu [15] selected external factors such as regional economic strength, policy environment, scientific research environment, educational investment strength and internal factors such as research talents and teachers' title structure in universities, and the regression results showed that regional scientific research environment, teachers' title structure and educational investment strength have a significant impact on scientific research efficiency. Wang and Jiang [16] selected economic advantage, location advantage, government support and R&D foundation as the influencing factors, and found that GDP per capita has a significant negative impact on the efficiency of scientific and technological innovation in universities.

Foreign research on scientific and technological innovation in universities can be broadly categorized into two types, one of which is to study the relationship between universities and enterprises or regional innovation. For example, Cowan and Zinovyeva [17] study the impact of universities on regional innovation, and the results show that industry-academia cooperation and university research activities can promote regional innovation. Ferguson and Fernandez [18] study the role of universities in the innovation system, emphasizing the importance of the cooperation between universities, enterprises and governments. María and Oscar [19] study the impact of technology transfer from universities on firms' innovation and conclude that there is a positive spillover effect of technology transfer from universities, which can promote the improvement of R&D capabilities within firms.

Another category is to study the research activities of universities themselves, such as Chavas et al., [20] used the DEA model to study the R&D activities of research universities in the U.S., and found that there are significant economies of scale between the papers and patents produced by universities. Guccio et al., [21] evaluate the teaching and scientific research efficiency of public universities in Italy, and analyze the relationship between the two as well as the convergence, and the study shows that research performance is the main driver of overall efficiency improvement. Boon and Andrew [22] constructed quantitative and qualitative oriented two-stage network DEA models to evaluate the research efficiency of Australian universities, and concluded that the evaluation results of the traditional DEA model were high, and that the efficiency level of the R&D stage was better than that of the funding application stage.

Geraint and Jill [23] used innovation income as a measure of research activities to study the cost and efficiency of UK universities. Shamohammadi et al., [24] categorized Korean private universities into teaching-oriented, research-oriented, and teaching-research-oriented based on the efficiency model, and used a two-stage network DEA model to study the changes in their efficiency, concluding that improving the quantity and quality of research outputs can contribute to the efficiency of universities. Mammadov and Aypay [25] evaluated the efficiency of Turkish research universities using the number of program members and the number of research projects as input variables, and the citation rate of the papers, the ratio of the income of the research projects to the total appropriation, and the ratio of the Ph.D. students to the number of Ph.D. projects as output variables.

The research of domestic scholars to evaluate the efficiency of scientific and technological innovation in colleges and universities can be mainly categorized into three types according to the different decision-making units.

The first category studies the innovation efficiency of certain types of key colleges and universities, such as Fu et al., [26], Jiang [27], Hu and Wang [28] with China's "985 project" colleges and universities as the object of the study, Wang and Wang [29], Zhu et al., [30], Li et al., [31] take "first-class university" type universities as the research objects, and Wang et al., [32] and Gao et al., [33] take ministry-affiliated colleges and universities as the research object, and measure the efficiency of their scientific and technological innovation respectively.

The second category studies the innovation efficiency of colleges and universities in a certain region, such as Wang [34], Su et al., [35], Song and Zou [36], and Zhao [37] take colleges and universities in Jiangsu, Zhejiang, Hubei and Shandong provinces as the research objects, and analyze the level and the differences in their scientific and technological innovation efficiency.

The third category compares the innovation efficiency of colleges and universities in different regions, for example, Zhang and Shang [38] used the Malmquist index to study the changes in the research and innovation efficiency of colleges and universities in China's provinces and municipalities, and found that the growth of total factor productivity was higher in the eastern region. Wang et al., [16] analyze the differences in the innovation efficiency of universities in different regions from the perspective of eight economic regions, and the results of the study show that the innovation efficiency of the western region was higher during the 12th Five-Year Plan period. Su and Liu [39], Zhang et al., [40] evaluated the input and output efficiency of scientific and technological innovation of colleges and universities in 31 provinces (autonomous regions and municipalities directly under the central government) under the background of the construction of "double first-class", and concluded that the research efficiency of colleges and universities in the western region is higher than that in the central and eastern regions.

Domestic scholars mainly use non-parametric methods to evaluate the efficiency of scientific and technological innovation in colleges and universities. In this case, Shen and Gong [41] used a three-stage DEA model to analyze the differences in the efficiency of scientific and technological innovation in universities in different provinces of China in order to eliminate the influence of environmental factors and random errors. Chen et al., [42] used the DEA window analysis model to dynamically analyze the research efficiency of universities in different regions of China and found that the level of efficiency decreases with increasing window width. Ni [43] comprehensively considered the lag, accuracy and preference of efficiency evaluation and used the lagged non-radial super-efficiency DEA model to measure the research efficiency of universities.

Qin et al., [13] divided the innovation process into three stages: knowledge acquisition, technological innovation and value transformation, and used the network SBM model based on slack variables to evaluate the efficiency of scientific research results transformation in China's interprovincial universities. Liu and Gong [44] used Bootstrap-DEA method to study the regional differences in the research efficiency of China's universities, and the statistical test results showed that the measure was more credible. Luo et al., [45] regarded technological innovation as a two-stage process of scientific and technological output and scientific and technological transformation, and used the DEA-Malmquist method to measure and analyze the innovation efficiency of universities in Jiangsu Province.

## 3. Materials and methods

### 3.1. Study design

This study aims to assess the efficiency of scientific and technological innovation (STI) activities in Shandong Vocational and Technical University of International Studies over a ten-year period, from 2013 to 2022. The primary objective is to evaluate the university's STI efficiency, considering both the knowledge innovation and results transformation stages of the innovation process. This will allow us to analyze how the university's research outputs (publications, patents, etc.) have contributed to its overall scientific and technological advancement.

The study employs a two-stage Data Envelopment Analysis (DEA) model, which is a well-established method for evaluating the efficiency of decision-making units (DMUs) in multi-input, multi-output settings. Specifically, the study evaluates the efficiency of STI activities at the university using input indicators such as R&D personnel and expenditures, and output indicators like academic papers, patents, and income from technology transfer. The two-stage approach allows for the separation of the innovation process into two distinct phases: knowledge innovation and results transformation, both of which contribute to the overall efficiency of the institution.

Hypotheses tested include the significant improvement of STI efficiency over the past decade, the positive correlation between the efficiency of the knowledge innovation stage and the results transformation stage, the positive impact of increasing R&D investment and the introduction of high-level research talents on STI efficiency, and the significant influence of the commercialization of research outputs, such as patents and technology transfers, on overall STI efficiency.

### 3.2. Data collection

This study evaluates the efficiency of scientific and technological innovation (STI) activities at Shandong Vocational and Technical University of International Studies (SVTUIS) from 2013 to 2022. The data, which were collected from various administrative and research departments such as the Research Office, Financial Department, and the Office of Academic Affairs, include both input indicators, such as resources allocated to R&D activities, and output indicators, like research results and their economic value. The ten-year period from 2013 to 2022 was chosen to provide a comprehensive overview of the university's STI activities and observe trends over time, with data for each year collected from official university records to ensure consistency and accuracy. The specific data collected for the evaluation encompass input indicators like R&D full-time personnel, R&D expenditure, laboratory base inputs, R&D FTE conversion, and gross salary of researchers, all of which were sourced from respective university departments. Output indicators include academic papers published, patents granted, and graduates, with additional input indicators for stage two focusing on results transformation, such as new technology R&D and transformation costs, new full-time researchers, total payroll for new personnel, and other costs associated with collaboration and resource sharing. Final output indicators consist of academic papers published in high-impact journals, patents obtained, scientific and technical publications points, income from technology transfer, research awards, and research waste or loss. Once collected, the data were normalized to account for differences in scale between the input and output indicators, with a specific transformation applied to negative values of research waste to make them positive for inclusion in the analysis. This normalization process ensures comparability and avoids biases in the DEA evaluation. The data sources include official reports from SVTUIS (2013–2022), the university's research office for academic papers and patents, the financial

department for expenditure data and payroll records, the HR department for personnel data, the Intellectual Property Office for patent and technology transfer data, and the student affairs office for graduation data.

## 3.3. Data preprocessing and normalization

Before applying the DEA model, the raw data were normalized to ensure comparability across different years. For the normalization process, the values from 2022 were used as the reference year, multiplying these values by a factor of 1.1 to ensure that the data across all years fall within a common scale.

## 4. Construction of evaluation system for scientific and technological innovation efficiency in universities

### 4.1. Selection of indicators

The scientific and technological innovation activities of colleges and universities are complex systems with multiple inputs, outputs and processes, which are affected by a variety of factors. In this study, a comprehensive set of input and output indicators was selected to evaluate the efficiency of scientific and technological innovation (STI) activities at Shandong Vocational and Technical University of International Studies (SVTUIS). The indicators were chosen based on their relevance to the innovation process, their ability to reflect the effectiveness of research activities, and their availability for the study period (2013–2022). The indicators used in this study are organized into input indicators, intermediate output indicators, and final output indicators (For specific indicator names, see Table 1).

 **4.1.1. Input indicators. 4.1.1.1. R&D full-time personnel ($X_{1j}$):** This indicator measures the total number of full-time personnel involved in scientific research, technology development, and technological services at the university. It includes researchers, R&D staff, and personnel dedicated to research administration and project management. This input is crucial

**Table 1. Evaluation index system of scientific and technological innovation efficiency of universities.**

| Indicator groups | Indicator names | Notation |
|---|---|---|
| Initial input indicators | R&D Full-time Personnel | $X_{1j}$ |
| | R&D Expenditure | $X_{2j}$ |
| | Laboratory Base Inputs | $X_{3j}$ |
| | R&D FTE conversion (person-years) | $X_{4j}$ |
| | Gross salary of researchers | $X_{5j}$ |
| Intermediate output indicators | Number of Academic Papers Published | $Z_{1j}$ |
| | Number of Patents Granted | $Z_{2j}$ |
| | Number of students graduated | $Z_{3j}$ |
| Additional input indicators | New Technology R&D and Transformation Costs ($ million) | $H_{1j}$ |
| | New full-time researchers (person-years) | $H_{2j}$ |
| | Total payroll for new personnel ($ million) | $H_{3j}$ |
| | Other costs to be increased or shared ($ million) | $H_{4j}$ |
| Final output indicators | Conversion score for academic paper publications | $Y_{1j}$ |
| | Number of patents obtained | $Y_{2j}$ |
| | Points for scientific and technical publications | $Y_{3j}$ |
| | Income from Technology Transfer | $Y_{4j}$ |
| | Research Awards | $Y_{5j}$ |
| | Research waste or loss ($ million) | $Y_{6j}$ |

because the human resources in R&D activities are a direct determinant of the capacity for scientific innovation and output.

**4.1.1.2. R&D expenditure ($X_{2j}$):** This indicator represents the total annual internal expenditure dedicated to R&D activities. R&D expenditure includes the funding for research projects, laboratory operations, and technological services. It is an essential measure of the financial investment made in scientific and technological innovation.

**4.1.1.3. Laboratory base inputs ($X_{3j}$):** This indicator refers to the share of resources allocated to research infrastructure, including the maintenance and upgrading of laboratories, research equipment, and technical facilities. A strong laboratory base is vital for fostering innovation and supporting research projects.

**4.1.1.4. R&D FTE (full-time equivalent) conversion ($X_{4j}$):** This indicator reflects the total amount of labor devoted to R&D activities, measured in full-time equivalent (FTE) years. It is calculated by converting the time spent by all researchers into person-years. This input is essential to assess the extent of human capital applied to scientific research.

**4.1.1.5. Gross salary of researchers ($X_{5j}$):** This indicator represents the total salary expenses for R&D personnel at the university. It provides insight into the university's investment in human capital for research and innovation. Higher salary expenditures are often associated with attracting high-caliber researchers and fostering a competitive research environment.

 **4.1.2. Intermediate output indicators. 4.1.2.1. Number of academic papers published ($Z_{1j}$):** This indicator measures the total number of academic papers published by university researchers in indexed journals. Publications are a key outcome of scientific research and indicate the university's contribution to advancing knowledge in various fields. Papers published in journals such as those indexed in the Chinese Social Sciences Citation Index (CSSCI) are given particular importance in this study.

**4.1.2.2. Number of patents granted ($Z_{2j}$):** This indicator tracks the number of patents granted to university researchers, representing the commercialization potential of research outputs. Patents are a key measure of the university's capacity to translate research into tangible innovations with potential economic value.

**4.1.2.3. Number of graduates ($Z_{3j}$):** This indicator measures the number of students graduating from programs directly related to R&D or science and technology. Graduates serve as a measure of the university's ability to cultivate a skilled workforce that can contribute to the development of new technologies and innovations.

 **4.1.3. Final output indicators (results transformation stage). 4.1.3.1. Conversion Score for Academic Paper Publications ($Y_{1j}$):** This indicator measures the academic value generated by the publication of scientific papers. It takes into account the quality, impact, and citation frequency of the published papers. Higher scores are assigned to papers published in higher-impact journals or those that are widely cited by the academic community.

**4.1.3.2. Number of Patents Obtained ($Y_{2j}$):** This final output indicator measures the number of patents granted to the university's R&D teams during the year. Patents are an important metric for the successful transformation of research into practical, commercially viable technologies.

**4.1.3.3. Points for Scientific and Technical Publications ($Y_{3j}$):** This indicator measures the impact of scientific and technical publications based on their contribution to advancing scientific knowledge and technology. The scoring system accounts for the journal's quality and the citation rate of published works.

**4.1.3.4. Income from Technology Transfer ($Y_{4j}$):** This indicator measures the financial income generated from the transfer of technology, including royalties and licensing fees. It reflects the economic impact of the university's R&D outputs and their successful commercialization.

**4.1.3.5. Research Awards ($Y_{5j}$):** This indicator tracks the number of research awards received by university researchers. It includes national and provincial research awards that acknowledge the scientific and technological achievements of the university. Higher awards indicate more significant contributions to scientific and technological progress.

**4.1.3.6. Research Waste or Loss ($Y_{6j}$):** This indicator measures the financial loss or waste incurred in R&D activities, including failed research projects and unutilized research outputs. Negative values in this category reflect inefficiencies

in the R&D process, and the goal is to minimize this output. In the model, research waste was treated as an undesirable variable, with appropriate adjustments made for its negative values (as detailed in the methodology section).

These indicators were selected to cover all aspects of the scientific and technological innovation process, from human and financial inputs to the outputs in terms of academic contributions and commercialization of research. The use of both input and output indicators ensures a comprehensive evaluation of the university's STI efficiency and helps identify areas for improvement in resource allocation, research performance, and technology commercialization.

## 4.2. Model building

Science and technology innovation activities in universities is a multi-stage value chain transfer process from innovation factor input to innovation product output, including both the use of innovation resources to produce knowledge results and the realization of the value of knowledge results, both of which are indispensable. Therefore, this paper divides the university science and technology innovation system into two subsystems: knowledge innovation and results transformation, and analyze the development of science and technology innovation in China's regional universities from the perspective of innovation value chain point of view. The two-stage network DEA model is often used to measure the efficiency value of the innovation system as a whole and each subsystem (Su and Liu, [39]2020), but it only considers the impact of the initial inputs on the first stage, ignoring the allocation of innovation resource inputs in each sub-process. In this paper, taking into account the issues of intermediate output reinvestment, allocation of initial inputs among sub-systems, and new inputs in the second stage, we proposed the following two-stage production system structure conceptual model (The model is shown in Fig 2.) based on the model presented by Chen and Guan [46].

This figure depicts the structure of the scientific and technological innovation (STI) system in higher education institutions, divided into two phases: the knowledge innovation phase and the transformation phase. In the knowledge innovation phase, inputs ($X_{ij}$) are allocated through procedure 1 ($\alpha_1 X_{ij}, (1-\alpha)X_{ij}$) to generate intermediate outputs ($Z_{ij}$). These intermediate outputs, together with additional resources ($H_{ij}$), serve as inputs for the transformation phase, where procedure 2 processes them into final outputs ($Y_{ij}$). The model highlights the sequential and interconnected nature of innovation and transformation within the STI system in universities.

In the process of university STI value chain estimation, we assume that there are $n$ decision-making units (DMUs), i.e., 15 vocational undergraduate colleges and universities selected in this paper, Each $DMU_j$ (j =1,2,...,n) has $m$ initial inputs $X_{ij}$ (i=1,2,...,m), $q$ intermediate products $Z_{pj}$ (p =1,2,...,q), $g$ additional inputs H (h=1,2,...,g), and $s$ final outputs $Y_{rj}$ (r=1,2,...,s). The initial input $X_{ij}$ does not act exclusively in the first stage, but is consumed jointly by both stages in some proportion, and the allocation varies across decision units. Assuming that the proportions of the initial inputs allocated to the first and second stages are $\alpha_i$ and $(1-\alpha_i)$ respectively ($0 < \alpha_i \leq 1$), the discretionary inputs of the first stage are $\alpha_i X_{ij}$ and the discretionary inputs of the second stage are $(1-\alpha_i)X_{ij}$, and the weighting structure of the two portions of the initial inputs in their respective processes are represented by the decision variables $v_i^1$ and $v_i^2$, respectively. The intermediate product $Z_{pj}$ is used as both the output of the first stage and the input of the second stage in the overall system, and the weight structure of the intermediate product in the output of the first stage and the input of the second stage are represented by the decision variables $\omega_p^1$ and $\omega_p^2$, respectively. In addition, the decision variable $f_h$ represents the weighting structure of the new inputs and the decision variable $u_r$ represents the weighting structure of the final outputs.

Therefore, the overall input of the knowledge innovation stage of the regional universities in China is $\sum_{i=1}^{m} v_i^1 \alpha_i X_{ij}$ and the overall output is $\sum_{n=1}^{q} \omega_p^1 Z_{pi}$; the overall input of the result transformation stage is $\sum_{i=1}^{m} v_i^2 (1-\alpha_i)X_{ij} + \sum_{n=1}^{q} \omega_p^2 Z_{pi} + \sum_{h=1}^{g} f_h H_{hj}$ and the overall output is $\sum_{r=1}^{s} u_r Y_{rj}$. Under the output orientation, the overall technical efficiency of any decision unit to be evaluated $DMU_k$ can be defined as:

$$E_k = \max \frac{\sum_{p=1}^{q} \omega_p^1 Z_{pk} + \sum_{r=1}^{s} u_r Y_{rk}}{\sum_{i=1}^{m} v_i^1 \alpha_i X_{ik} + \sum_{i=1}^{m} v_i^2 (1-\alpha_i)X_{ik} + \sum_{p=1}^{q} \omega_p^2 Z_{pk} + \sum_{h=1}^{g} f_h H_{hk}}$$

(1)

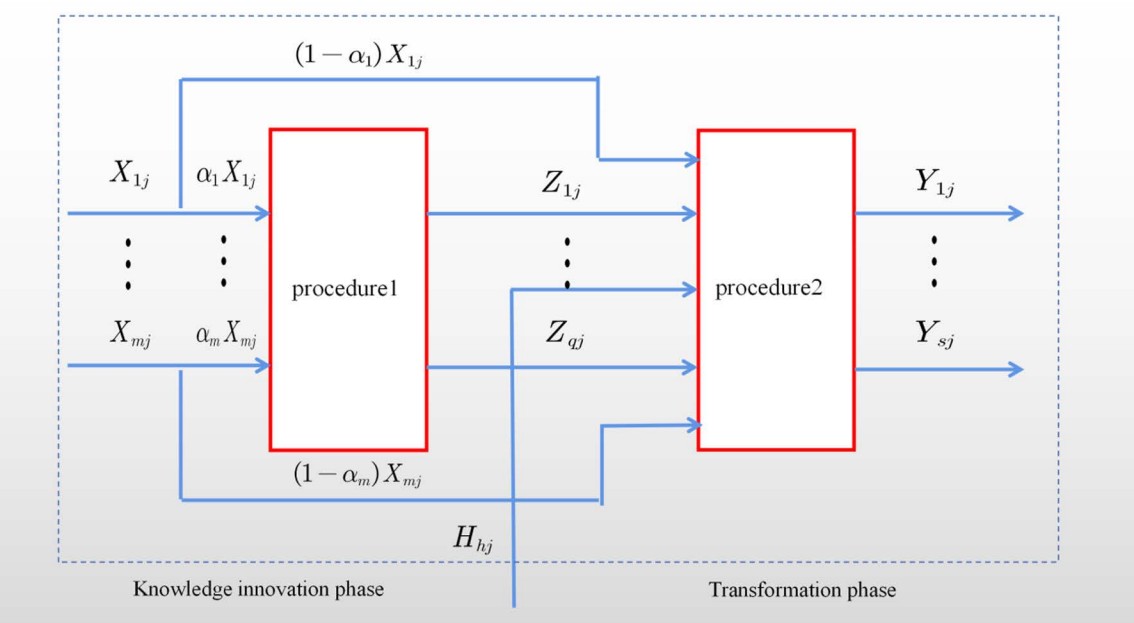

**Fig 2. Structure of the STI System in Higher Education Institutions.**

$$s.t. \begin{cases} \dfrac{\sum_{p=1}^{q} \omega_p^1 Z_{pj} + \sum_{r=1}^{s} u_r Y_{rj}}{\sum_{i=1}^{m} v_i^1 \alpha_i X_{ij} + \sum_{i=1}^{m} v_i^2 (1-\alpha_i) X_{ij} + \sum_{p=1}^{q} \omega_p^2 Z_{pj} + \sum_{h=1}^{g} f_h H_{hj}} \leq 1 \\[2ex] \dfrac{\sum_{p=1}^{q} \omega_p^1 Z_{pj}}{\sum_{i=1}^{m} v_i^1 \alpha_i X_{ij}} \leq 1 \\[2ex] \dfrac{\sum_{r=1}^{s} u_r Y_{rj}}{\sum_{i=1}^{m} v_i^2 (1-\alpha_i) X_{ij} + \sum_{p=1}^{q} \omega_p^2 Z_{pj} + \sum_{h=1}^{g} f_h H_{hj}} \leq 1 \\[1ex] v_i^1, v_i^2, \omega_p^1, \omega_p^2, f_h, u_r \geq 0; i = 1, 2, \cdots, n; j = 1, 2, \cdots, n \end{cases}$$

Let: $t = 1/\sum_{i=1}^{m} v_i^1 \alpha_i X_{ik} + \sum_{i=1}^{m} v_i^2 (1-\alpha_i) X_{ik} + \sum_{n=1}^{q} \omega_p^2 Z_{pk} + \sum_{h=1}^{g} f_h H_{hk}$, based on Charnes-Cooper transformation, the mathematical planning (4.1) can be simplified as:

$$E_k = \max \sum_{p=1}^{q} W_p^q Z_{pk} + \sum_{r=1}^{s} U_r Y_{rk} \tag{2}$$

$$s.t. \begin{cases} \sum_{i=1}^{m} V_i^1 \alpha_i X_{ik} + \sum_{i=1}^{m} V_i^2 (1-\alpha_i) X_{ik} + \sum_{p=1}^{q} W_p^2 Z_{pk} + \sum_{h=1}^{g} F_h H_{hk} = 1 \\[1ex] \sum_{i=1}^{m} V_i^1 \alpha_i X_{xj} - \sum_{p=1}^{q} W_p^1 Z_{pj} \geq 0 \\[1ex] \sum_{i=1}^{m} V_i^2 (1-\alpha_i) X_{ij} + \sum_{p=1}^{q} W_p^2 Z_{pj} + \sum_{h=1}^{g} F_h H_{hk} - \sum_{r=1}^{s} U_r Y_{rj} \geq 0 \\[1ex] V_i^1, V_i^2, W_p^1, W_p^2, F_h, U_r \geq \varepsilon; i = 1, 2, \cdots, m; j = 1, 2, \cdots, n \end{cases}$$

Where the decision variables $V_i^1 = t v_i^1$, $V_i^2 = t v_i^2$, $W_p^1 = t \omega_p^1$, $W_p^2 = t \omega_p^2$, $F_h = t f_h$, $U_r = t u_r$, to prevent the optimal value of the decision variables from 0 during the solution process, the lower bound of the decision variables is set to $\varepsilon$, i.e.,

non-Archimedean infinitesimal. In order to facilitate the solution, let $\pi_i^1 = V_i^1 \alpha_i$, $\pi_i^2 = V_i^2 \alpha_i$, then the above nonlinear planning can be transformed into an equivalent linear planning:

$$E_k = \max \sum_{p=1}^{q} W_p^1 Z_{pk} + \sum_{r=1}^{s} U_r Y_{rk} \tag{3}$$

$$s.t. \begin{cases} \sum_{i=1}^{m} \pi_i^1 X_{ik} + \sum_{i=1}^{m} V_i^2 X_{ik} - \sum_{i=1}^{m} \pi_i^2 X_{ik} + \sum_{p=1}^{q} W_p^2 Z_{pk} + \sum_{h=1}^{g} F_h H_{hk} = 1 \\ \sum_{i=1}^{m} \pi_i^1 X_{ij} - \sum_{p=1}^{q} W_p^1 Z_{pj} \geq 0 \\ \sum_{i=1}^{m} V_i^2 X_{ij} - \sum_{i=1}^{m} \pi_i^2 X_{ij} + \sum_{p=1}^{q} W_p^2 Z_{pj} + \sum_{h=1}^{s} F_h H_{hk} - \sum_{r=1}^{s} U_r Y_{rj} \geq 0 \\ \pi_i^1, \pi_i^2, W_p^1, W_p^2, F_h, U_r \geq \varepsilon; i = 1, 2, \cdots, m; j = 1, 2, \cdots, n \end{cases}$$

This linear programming is an input-oriented model for measuring the overall technical efficiency of the decision-making unit, from which the optimal combination of solutions for the decision variables $\alpha_i$, $V_i^1$, $V_i^2$, $W_p^1$, $W_p^2$, $F_h$ and $U_r$ can be obtained, thus allowing the calculation of the values of the technical efficiency value of the first stage and the second stage within the system, as shown in equation (4).

$$E_{ik}^1 = \frac{\sum_{p=1}^{q} W_p^1 Z_{pk}}{\sum_{i=1}^{m} \alpha V_k^1 X_{ik}} \tag{4}$$

Using the output efficiency of the first stage as the input factor of the second stage, the DEA model is used to calculate the output efficiency of the second stage again. If we use $E_{ik}^2$ the second stage, the STI (science, technology and innovation) efficiency of indicator i in period k, the expression is as follows:

$$E_{ik}^2 = \frac{\sum_{r=1}^{s} U_r Y_{rk}}{\sum_{i=1}^{m} (1-\alpha) V_i^2 X_{ik} + \sum_{p=1}^{q} W_p^2 Z_{pk} + \sum_{h=1}^{g} F_h H_{hk}} \tag{5}$$

## 5. Case studies and Results

Shandong Vocational and Technical University of International Studies, founded in 2005, is an undergraduate vocational and technical university approved by the Ministry of Education, with a degree-granting right at the undergraduate level, and is actively declaring the right to grant professional master's degrees. It has been developing rapidly in the past ten years, and according to the statistics of Shandong Vocational and Technical University of International Studies, the basic information for the ten-year period from 2013 to 2022 is detailed in Table 2. In this study, the R software (version 3.6.1) and its DEA package were used for STI (science, technology and innovation) efficiency assessment. Through this method, the input-output efficiency of STI in vocational undergraduate institutions was assessed and the trend of efficiency improvement was predicted and analyzed.

In order to improve the evaluation effect, it is necessary to normalize the above indicators; 18 evaluation indicators 1.1 times in 2022 are selected as the maximum value, and the determined maximum value is used to normalize the evaluation indicators. The results after the normalization process are presented in Table 3.

**Table 2. Basic data for evaluation of scientific and technological innovation efficiency of Shandong Vocational and Technical University of International Studies.**

| notation | 2013 | 2014 | 2015 | 2016 | 2017 | 2018 | 2019 | 2020 | 2021 | 2022 |
|---|---|---|---|---|---|---|---|---|---|---|
| $X_{1j}$ | 1128 | 1185 | 1236 | 1328 | 1438 | 1528 | 1639 | 1727 | 1826 | 1962 |
| $X_{2j}$ | 3516 | 3658 | 3701 | 3852 | 3974 | 4028 | 4172 | 4238 | 4328 | 4439 |
| $X_{3j}$ | 685 | 692 | 698 | 702 | 713 | 724 | 732 | 746 | 756 | 768 |
| $X_{4j}$ | 560 | 571 | 579 | 583 | 595 | 605 | 618 | 628 | 637 | 646 |
| $X_{5j}$ | 4260 | 4370 | 4420 | 4480 | 4510 | 4580 | 4650 | 4760 | 4810 | 4900 |
| $Z_{1j}$ | 85 | 89 | 95 | 102 | 113 | 121 | 128 | 132 | 138 | 142 |
| $Z_{2j}$ | 75 | 80 | 85 | 89 | 92 | 97 | 102 | 109 | 112 | 121 |
| $Z_{3j}$ | 2100 | 2300 | 2400 | 3100 | 3500 | 4100 | 4600 | 4800 | 5100 | 5400 |
| $H_{1j}$ | 125 | 131 | 138 | 142 | 149 | 152 | 158 | 162 | 168 | 172 |
| $H_{2j}$ | 112 | 128 | 131 | 138 | 141 | 149 | 153 | 157 | 162 | 167 |
| $H_{3j}$ | 380 | 410 | 440 | 470 | 510 | 550 | 580 | 610 | 640 | 660 |
| $H_{4j}$ | 139 | 145 | 151 | 158 | 161 | 168 | 172 | 181 | 185 | 208 |
| $Y_{1j}$ | 680 | 820 | 910 | 950 | 980 | 1050 | 1080 | 1120 | 1160 | 1200 |
| $Y_{2j}$ | 180 | 220 | 250 | 280 | 320 | 350 | 380 | 410 | 480 | 510 |
| $Y_{3j}$ | 178 | 189 | 196 | 205 | 212 | 225 | 236 | 247 | 256 | 268 |
| $Y_{4j}$ | 205 | 218 | 226 | 231 | 242 | 248 | 252 | 259 | 265 | 276 |
| $Y_{5j}$ | 256 | 267 | 278 | 289 | 305 | 310 | 320 | 330 | 340 | 350 |
| $Y_{6j}$ | -126 | -131 | -138 | -146 | -150 | -156 | -168 | -175 | -181 | -185 |

**Table 3. Normalized results of evaluation indexes of scientific and technological innovation efficiency of Shandong Vocational and Technical University of International Studies.**

| notation | 2013 | 2014 | 2015 | 2016 | 2017 | 2018 | 2019 | 2020 | 2021 | 2022 |
|---|---|---|---|---|---|---|---|---|---|---|
| $X_{1j}$ | 0.5227 | 0.5491 | 0.5727 | 0.6153 | 0.6663 | 0.7080 | 0.7594 | 0.8002 | 0.8461 | 0.9091 |
| $X_{2j}$ | 0.7201 | 0.7491 | 0.7580 | 0.7889 | 0.8139 | 0.8249 | 0.8544 | 0.8679 | 0.8864 | 0.9091 |
| $X_{3j}$ | 0.8108 | 0.8191 | 0.8262 | 0.8310 | 0.8440 | 0.8570 | 0.8665 | 0.8830 | 0.8949 | 0.9091 |
| $X_{4j}$ | 0.7881 | 0.8035 | 0.8148 | 0.8204 | 0.8373 | 0.8514 | 0.8697 | 0.8838 | 0.8964 | 0.9091 |
| $X_{5j}$ | 0.7904 | 0.8108 | 0.8200 | 0.8312 | 0.8367 | 0.8497 | 0.8627 | 0.8831 | 0.8924 | 0.9091 |
| $Z_{1j}$ | 0.5442 | 0.5698 | 0.6082 | 0.6530 | 0.7234 | 0.7746 | 0.8195 | 0.8451 | 0.8835 | 0.9091 |
| $Z_{2j}$ | 0.5635 | 0.6011 | 0.6386 | 0.6687 | 0.6912 | 0.7288 | 0.7663 | 0.8189 | 0.8415 | 0.9091 |
| $Z_{3j}$ | 0.3535 | 0.3872 | 0.4040 | 0.5219 | 0.5892 | 0.6902 | 0.7744 | 0.8081 | 0.8586 | 0.9091 |
| $H_{1j}$ | 0.6607 | 0.6924 | 0.7294 | 0.7505 | 0.7875 | 0.8034 | 0.8351 | 0.8562 | 0.8879 | 0.9091 |
| $H_{2j}$ | 0.6097 | 0.6968 | 0.7131 | 0.7512 | 0.7676 | 0.8111 | 0.8329 | 0.8547 | 0.8819 | 0.9091 |
| $H_{3j}$ | 0.5234 | 0.5647 | 0.6061 | 0.6474 | 0.7025 | 0.7576 | 0.7989 | 0.8402 | 0.8815 | 0.9091 |
| $H_{4j}$ | 0.6075 | 0.6337 | 0.6600 | 0.6906 | 0.7037 | 0.7343 | 0.7517 | 0.7911 | 0.8086 | 0.9091 |
| $Y_{1j}$ | 0.5152 | 0.6212 | 0.6894 | 0.7197 | 0.7424 | 0.7955 | 0.8182 | 0.8485 | 0.8788 | 0.9091 |
| $Y_{2j}$ | 0.3209 | 0.3922 | 0.4456 | 0.4991 | 0.5704 | 0.6239 | 0.6774 | 0.7308 | 0.8556 | 0.9091 |
| $Y_{3j}$ | 0.6038 | 0.6411 | 0.6649 | 0.6954 | 0.7191 | 0.7632 | 0.8005 | 0.8379 | 0.8684 | 0.9091 |
| $Y_{4j}$ | 0.6752 | 0.7181 | 0.7444 | 0.7609 | 0.7971 | 0.8169 | 0.8300 | 0.8531 | 0.8729 | 0.9091 |
| $Y_{5j}$ | 0.6649 | 0.6935 | 0.7221 | 0.7506 | 0.7922 | 0.8052 | 0.8312 | 0.8571 | 0.8831 | 0.9091 |
| $Y_{6j}$ | 0.6192 | 0.6437 | 0.6781 | 0.7174 | 0.7371 | 0.7666 | 0.8256 | 0.8600 | 0.8894 | 0.9091 |

On the basis of the above data, the technical efficiency of the first stage is calculated using formula (4). Then the output efficiency of the first stage is used as the input element of the second stage, and formula (5) is used to calculate the efficiency of the scientific and technological innovation of Shandong Vocational and Technical University of International Studies for the period of 2013–2022 The results are detailed in Table 4.

According to the data in Table 4, the results of the evaluation of the efficiency of science and technology innovation in Shandong Vocational and Technical University of International Studies can be derived by weighting. We divide the efficiency of science and technology innovation into four levels according to the average efficiency score, and the specific division criteria are as follows:

Level I (Highest Efficiency): This level includes cases with efficiency scores above 0.9 (inclusive). Years belonging to this level indicate that the university's S&T innovation efficiency is close to or has reached the optimal level in that period, and the use of scientific research resources and the output of scientific and technological achievements have reached a state of high coordination and efficient utilization.

Level II (High Efficiency): Cases with efficiency scores between 0.8 (inclusive) and 0.899 are categorized into this level. It indicates that the university is performing well in STI and utilizing resources efficiently, but there is still room for improvement to achieve optimal efficiency.

Level III (Moderate Efficiency): The efficiency score of this level ranges from 0.7 (inclusive) to 0.799, reflecting that the university's S&T and innovation activities at this stage have certain efficiency deficiencies, and need to further optimize the management of scientific research projects, the use of funds, and the allocation of talents.

Level IV (Low Efficiency): Efficiency scores below 0.7 are categorized in this level. It suggests that the university's S&T innovation is less efficient in this period, which may be due to improper allocation of resources, deficiencies in the research management system, or the influence of external environmental factors.

This figure illustrates the trends in scientific and technological innovation efficiency in higher education institutions from 2013 to 2022, including first-stage efficiency (blue line), second-stage efficiency (green line), and average efficiency (red line). The first-stage efficiency represents the knowledge innovation phase, while the second-stage efficiency corresponds to the transformation phase. Over the observed period, all three efficiency metrics show a steady increase, indicating continuous improvement in the capability of higher education institutions to transform research inputs into outputs effectively.

The study found that from 2013 to 2022, there is a significant increase in the efficiency of science and technology innovation in Shandong Vocational and Technical University of International Studies, with the efficiency value increasing from 0.6194 to 0.9091, as shown in Figure 3. This trend reflects the successful adjustment of the university in terms of STI management and strategic positioning. The main internal and external factors contributing to this change are analyzed below:

**Table 4. Evaluation results of scientific and technological innovation efficiency of Shandong Vocational and Technical University of International Studies.**

| Year | First Stage Efficiency | Second Stage Efficiency | average efficiency | Evaluation level |
|------|------------------------|-------------------------|--------------------|------------------|
| 2013 | 0.5635 | 0.6752 | 0.6194 | Level IV |
| 2014 | 0.6011 | 0.7181 | 0.6596 | Level IV |
| 2015 | 0.6386 | 0.7444 | 0.6915 | Level IV |
| 2016 | 0.6687 | 0.7609 | 0.7148 | Level III |
| 2017 | 0.7234 | 0.7971 | 0.7603 | Level III |
| 2018 | 0.7746 | 0.8169 | 0.7958 | Level III |
| 2019 | 0.8195 | 0.8300 | 0.8248 | Level II |
| 2020 | 0.8451 | 0.8531 | 0.8491 | Level II |
| 2021 | 0.8835 | 0.8729 | 0.8782 | Level II |
| 2022 | 0.9091 | 0.9091 | 0.9091 | Level I |

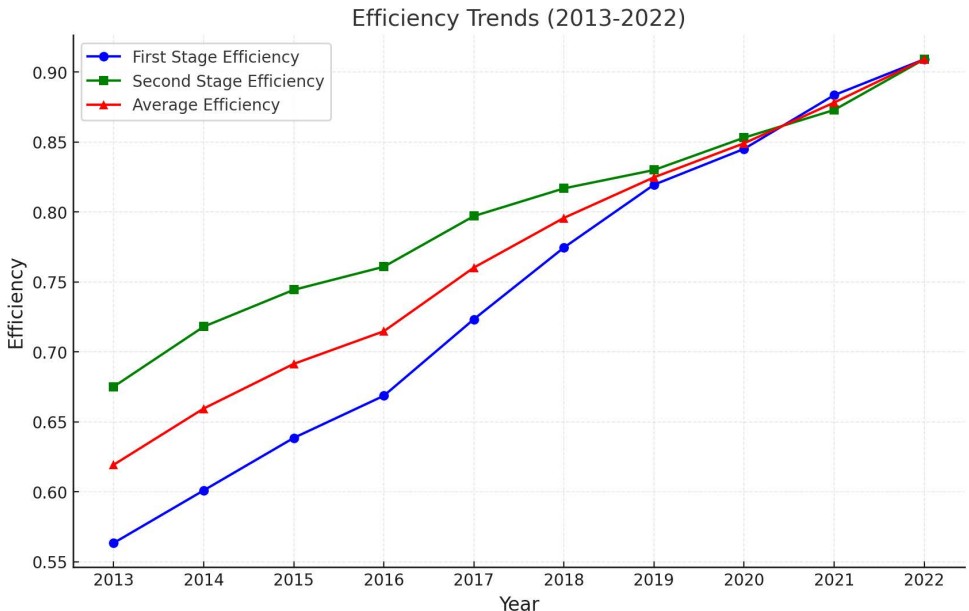

**Fig 3. Trend of Scientific and Technological Innovation Efficiency (2013-2022).**

5.1 The university has significantly improved the overall capability of its research team by introducing high-level research talents and strengthening the training of existing researchers. Quantitative evidence supporting this improvement includes a 15.4% growth in the number of full-time R&D personnel ($X_{4j}$), which increased from 560 in 2013–646 in 2022, demonstrating the university's efforts to strengthen its research team through targeted recruitment. Furthermore, the introduction of high-level talents has resulted in a substantial increase in research output, with academic papers published ($Z_{1j}$) rising by 67%, from 85 in 2013–142 in 2022, and patents granted ($Z_{2j}$) increasing by 61.3%, from 75 to 121 during the same period. Simultaneously, the university's R&D expenditure ($X_{5j}$) grew by 15%, from 426,000 yuan in 2013–490,000 yuan in 2022, enabling the recruitment of top-tier researchers and the initiation of more advanced research projects. These combined efforts indicate significant progress in enhancing the university's research team capabilities through strategic recruitment, increased output, and sustained investment.

5.2 The university's research efficiency has also improved significantly, as evidenced by multiple indicators. The DEA model reveals that the efficiency scores for the Knowledge Innovation Stage (Stage 1) improved from 0.5635 in 2013 to 0.9091 in 2022, reflecting a 61.3% increase and indicating an enhanced capability to convert research inputs into knowledge outputs. This improvement was supported by higher output efficiency, as shown by the increased number of academic papers ($Z_{1j}$) and patents ($Z_{2j}$), which reflect more effective utilization of research inputs. Additionally, the return on investment (ROI) for R&D expenditure improved by 25%, with patents per million yuan of R&D funding rising from 0.02 in 2013 to 0.025 in 2022. These improvements collectively demonstrate the university's significant progress in research efficiency, driven by stronger resource allocation and a more capable research team.

5.3 Moreover, the university has strengthened its collaboration with industry and government, leading to improved technology commercialization and increased research capacity. For instance, income from technology transfer ($Y_{4j}$) grew by 34.6%, rising from 205,000 yuan in 2013–276,000 yuan in 2022, which highlights the role of industry partnerships in commercializing research outcomes. Government financial support for technology transfer ($Y_{5j}$) also increased by 36.7%, from 256,000 yuan in 2013–350,000 yuan in 2022, further supporting R&D activities and commercialization

efforts. Additionally, the proportion of external funding in the university's R&D budget rose from 20% in 2013 to 28% in 2022, reflecting stronger collaborations with external stakeholders.

These factors, alongside the university's enhanced research team and improved efficiency, have contributed to its success in aligning resources and strategies with the demands of scientific and technological innovation.

## 6. Discussion

### 6.1 . Implications of Results

The results of this study provide critical insights into improving STI efficiency in vocational universities. The findings demonstrate a significant upward trend in STI efficiency at Shandong Vocational and Technical University of International Studies, increasing from 0.6194 in 2013 to 0.9091 in 2022. This aligns with previous research, such as Chen et al., [42], which emphasized that optimized funding and efficient resource allocation play a pivotal role in improving research productivity, and Abramo et al., [7], which highlighted the impact of high-performing researchers on overall institutional efficiency. The study underscores the importance of well-targeted investments in research and development (R&D), faculty expertise, and innovation strategies to enhance research productivity and commercialization outcomes.

The observed improvements suggest that STI efficiency contributes not only to the institutional development of vocational universities but also to their broader role in regional economic growth and innovation ecosystems. For example, vocational universities are uniquely positioned to address practical industry needs, and improved STI efficiency enables these institutions to produce tangible outcomes such as patents, technical solutions, and industry partnerships that directly benefit regional economies.

### 6.2. Factors Influencing Efficiency

The observed improvements in STI efficiency can be attributed to several key factors. First, the introduction of high-level research talents has been instrumental in driving innovation. Faculty with advanced expertise and strong research backgrounds have contributed to higher-quality outputs, including academic publications and patents.

This aligns with previous studies (e.g., Benedetto et al., [9]) that emphasize the positive correlation between researcher quality and institutional efficiency.

Second, increased R&D investment has provided the necessary financial foundation for enhancing STI outcomes. Targeted funding for research infrastructure, laboratories, and technology transfer initiatives has significantly improved resource utilization efficiency.

Third, the university's strategic focus on applied research and industry collaboration has played a critical role. By fostering partnerships with local industries, the university has successfully aligned its research activities with real-world applications, leading to the commercialization of outputs such as patents and technological innovations. This industry-academia synergy ensures that STI activities are demand-driven and impactful.

Finally, the university's internal institutional strategies, including improved project management, faculty development programs, and resource optimization, have enhanced the overall innovation ecosystem. These strategies address both the knowledge innovation and results transformation stages, ensuring efficient resource allocation throughout the innovation value chain.

### 6.3. Comparison with Other Universities

When compared to other vocational and technical universities in China or internationally, Shandong Vocational and Technical University of International Studies demonstrates a noteworthy improvement in STI efficiency. However, certain areas for further enhancement remain evident.

For instance, universities in more developed regions or countries may exhibit higher efficiency due to better access to funding, advanced infrastructure, and established collaborations with industry (Kolympiris & Klein, [10]). By benchmarking against these institutions, vocational universities like Shandong Vocational and Technical University of International Studies can identify replicable strategies and further optimize their STI processes.

### 6.4. Limitations of the Study

Despite its contributions, this study has certain limitations that must be acknowledged. First, the analysis focuses on a single university, which may limit the generalizability of the findings to other institutions, especially those in different regional or economic contexts. Future research should consider expanding the scope to include multiple vocational universities for a broader comparative analysis.

Second, the study relies on quantitative data and the DEA model, which assumes linear relationships between inputs and outputs. While DEA is a robust method for efficiency evaluation, it does not account for qualitative factors such as institutional culture, policy frameworks, or researcher motivation, which may also influence STI efficiency.

Third, potential data gaps may have impacted the accuracy of efficiency calculations. For example, intermediate outputs such as collaborative research projects or intangible innovation outcomes were not fully considered. Addressing these limitations in future studies through more comprehensive data collection and hybrid evaluation models (e.g., integrating qualitative analysis or dynamic DEA models) would improve the robustness of the results.

## 7. Conclusions and Proposing

Scientific research and innovation in vocational undergraduate colleges and universities is a key link in building a vertically integrated modern vocational education system and realizing the transformation of vocational education from "level" to "type", which is of great significance to the implementation of the innovation-driven development strategy and the acceleration of the formation of new productive forces. Under the perspective of innovation value chain, this paper decomposes the scientific and technological innovation process of colleges and universities into two sub-systems, namely, knowledge innovation and results transformation, and comprehensively considers the issues of intermediate output re-investment, initial input resource sharing, and new inputs in the second stage. It constructs the shared-input-type correlation DEA model with additional inputs, to measure the comprehensive efficiency of scientific and technological innovation of China's vocational undergraduate colleges and universities and the efficiency of the two sub-stages, as well as to assess the dynamic evolution status. The model analyzes and establishes an evaluation system for the dynamic evolution of the STI model and the synergistic effect of the two sub-stages.

7.1 It can be helpful to the efficient promotion of scientific and technological innovation activities in colleges and universities. By studying the efficiency of scientific and technological innovation in China's colleges and universities, identifying the loopholes in R&D input and output activities, and clarifying the important driving factors in the innovation process, our described approach can help colleges and universities to choose the right direction for scientific research improvement and formulate practical and effective measures in order to realize the optimal allocation of scientific research resources and improve the quality of scientific and technological output.

7.2 It is conducive to providing a basis for the formulation of relevant government policies. The good development of scientific research in colleges and universities depends on the government's policy guidance and financial support. Due to the differences in geographic location and resource endowment, there are gaps in the input of scientific research factors and the conditions of scientific research environment in colleges and universities in different regions, so a comparative analysis of the differences in the efficiency of scientific and technological innovation in colleges and universities in different regions will help the government formulate realistic policy programs according to the overall development of scientific and technological innovation and the differences in the region and equalize the development needs of the various regions.

7.3 It is conducive to promoting regional economic development. Science and technology innovation in colleges and universities is an important part of national and regional science and technology innovation, and its development status affects regional innovation capacity and socio-economic growth from a global and long-term perspective. This paper studies the efficiency of scientific and technological innovation in universities based on the perspective of innovation value chain, better connects the innovation activities of universities with the government and enterprises, so that universities can clearly define their role in the innovation system, thus further improving the level of cooperation between industry, academia and research, increasing the output of scientific and technological achievements, and providing support for the development of regional scientific and technological innovation.

## Author contributions

**Conceptualization:** Jun Zeng.

**Data curation:** Tao Sun.

**Funding acquisition:** Li Zhang.

**Investigation:** Li Zhang.

**Project administration:** Jun Zeng.

**Supervision:** Jun Zeng.

**Writing – original draft:** Tao Sun.

**Writing – review & editing:** Jun Zeng.

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
