## [Decision Letter · Decision Letter 0]

19 Nov 2024

Dear Dr. Zeng,

Thank you for submitting your manuscript to PLOS ONE. After careful consideration, we feel that it has merit but does not fully meet PLOS ONE’s publication criteria as it currently stands. Therefore, we invite you to submit a revised version of the manuscript that addresses the points raised during the review process.

We look forward to receiving your revised manuscript.

Kind regards,

Alexandre Morais Nunes, Ph.D.

Academic Editor

PLOS ONE

3. Please ensure that you include a title page within your main document. You should list all authors and all affiliations as per our author instructions and clearly indicate the corresponding author.

Reviewers' comments:

Reviewer's Responses to Questions

**Comments to the Author**

1. Is the manuscript technically sound, and do the data support the conclusions?

Reviewer #1: Yes

Reviewer #2: No

2. Has the statistical analysis been performed appropriately and rigorously?

Reviewer #1: Yes

Reviewer #2: No

3. Have the authors made all data underlying the findings in their manuscript fully available?

Reviewer #1: Yes

Reviewer #2: Yes

4. Is the manuscript presented in an intelligible fashion and written in standard English?

Reviewer #1: Yes

Reviewer #2: No

Reviewer #1: Overall evaluation

This paper presents detailed data from Shandong University of Vocational and Technical International Studies for the period 2013-2022. I find the subject matter of the paper valuable and have enjoyed the scientific approach.

The tables present basic data, normalised results and evaluation results of scientific and technological innovation efficiency. It is very valuable that the author analyses a 10-year data and shows the difference with the evaluation.

I can understand the use of modified titles for your paper, but you should take into account the journal guidelines and the reader's habits of browsing the research.

The author's focus on the evaluation system and showing that it works in evaluation is also understandable and deserves respect. However, we have come a long way in terms of access to data and the use of smart evaluation systems and the author should focus less on the evaluation system and more on what it evaluates and to what extent it explains the resulting change.

Readers and researchers will be interested in the section where the study states that from 2013 to 2022 there was a significant increase in the productivity of science and technology innovation at the university, revealing that the value of productivity increased. I would very much like this section to be discussed in connection with the findings.

The Abstract

Research has clear objective, but it could be more specific. Mentioning the goal of improving or evaluating the efficiency of innovation activities would provide better context. Explaining why the increase in research efficiency is important and how it impacts the institution, or the field of vocational education would add value. The abstract mentions proposing policy recommendations but does not provide any details. Including a brief mention of the type of recommendations (e.g., funding allocation, curriculum changes) would make the abstract more informative and appealing to readers. The abstract has enough space to add more detail about the policy recommendations and the significance of the findings without exceeding the limit.

Materials and Methods

The Materials and Methods section should provide enough detail to allow suitably skilled investigators to fully replicate your study. But this part of the article is hard to find and hard to follow. The relevant section should be structured by adding material and method section. For this reason, I have included the journal's article sections and expectations below my review.

Results, Discussion, Conclusions

These sections may all be separate, or may be combined to create a mixed Results/Discussion section (commonly labeled “Results and Discussion”) or a mixed Discussion/Conclusions section (commonly labeled “Discussion”). These sections may be further divided into subsections, each with a concise subheading, as appropriate. These sections have no word limit, but the language should be clear and concise.

Materials and Methods

Study Design: Please describe the overall design of the study, including the objectives and hypotheses.

Data Collection: Please give detail the data sources, including the period (2013-2022) and the specific data collected from Shandong Vocational and Technical University of International Studies.

Indicators Selection: Please list and describe the input and output indicators used in the study, such as R&D full-time personnel, R&D expenditure, number of published scientific writings, and patents granted.

Model Construction: Please explain the shared-input-type correlation DEA model and the two-stage network DEA model. Describe how intermediate output reinvestment, initial input resource sharing, and new inputs were factored into the model.

Data Analysis: Please give detail the analysis and explain the normalization process for the evaluation indicators

The manuscript does not have a clearly labeled “Results” or “Results and Discussion” section. However, the results and their discussion are embedded within the “Case Studies and Discussions” section. The manuscript should include a distinct “Results” or “Results and Discussion” section.

Reviewer #2: The authors have serious problems with references as many mistakes occur. The references must be profoundly revised. Similarly, there are many missing spaces or extra/double spaces in the text. Moreover, please avoid using very long paragraphs within the text. Try to keep all paragraphs within 10-12 lines for better readiness. Please revise the complete article or use a proofreading service. Besides, I have the following comments/recommendations:

Introduction

- Please add a reference related to the 18th CPC National Congress in the first paragraph.

- “…modernization country, and to accelerate…” Split the sentence into two here.

- “In December 2020, the Ministry of Education convened…” – reference is missing. The same in “In May 2019, the Ministry of Education announced the first 15 pilot…”

- Please rearrange chronologically the Introduction. The authors are jumping from September 2020, to December 2020, then to May 2019.

- “Scientific and technological innovation activities of colleges and universities…” – reference(s) is missing in this part. Moreover, you should add a figure to display this process.

- “Although the overall development level of scientific and technological innovation in colleges and universities in China has been significantly improved…” – how? Please, be more specific and provide some statistics, figures, etc., and support your statement with references.

- Further, the same, “there are still problems such as low utilization of innovation resources and unbalanced regional development” – references are missing.

- “Although the overall development level of scientific…” – this sentence has 13 lines! Please, revise this sentence and divide it into several shorter ones to improve the readiness and flow.

- “The main reasons for the inefficiency of innovation are clarified.” – what does it mean? Where?

Research on the influencing factors of scientific and technological innovation efficiency in universities

- This chapter is only one paragraph! Divide the text into several shorter and logical paragraphs.

Research on the evaluation of scientific and technological innovation efficiency in universities

- “…technological innovation efficiency, respectively.DIVIDE THE PARAGRAPH HERE The third category compares…”.

- It is not necessary to explain all forms of the DEA methodology used in the STI evaluation (CCR, BCC, two/three stages, MI and WA models. It is enough saying that the non-parametric approach based on the DEA methodology is used, and continue directly: “In this case, Shen and Gong (2013) used…”

Selection of indicators

- In table 1, please better divide the inputs, intermediates and outputs, use a simple line between each category. Right now one can think, for example, that the intermediates begin on the line next to Intermediate output indicators. The same with the Additional input indicators.

- The authors mention that “…the result transformation stage include part of the initial input resources”. – Which of the initial inputs are later partially used? It is not clear from Table 1.

- The authors refer to Table 4.1 and 4.2, but I do not see these tables in the article. Rather, Table 1, etc. In this case, Table 4.2 refers to Table 2? I am not sure about it.

- Could you please explain a bit how the Y6j Research waste output was treated in the model as it has negative values? Further, I do not understand how you transformed the negative values in Table 2 to positive normalized values in Table 3! Did you treat this variable as an undesirable variable? If yes, how exactly?

- Regarding the normalized values, why 1.2 of 2020 values was selected? Why not to use as a basis 2013 as 1.000 and then correspondingly normalize the rest of the years (such as GDP in prices of XXXX year.

Model building

- “ g additional inputs H (h=1,2,...,g), and g new inputs H (h=1,2,...,g). ...,q), g new inputs Hhj (h=1,2,...,g),” – first, I guess this description is an error. Second, does this description want to say that there are additional new inputs and shared inputs to Stage 2? If yes, use different parameters letter (for example, g additional inputs and r shared inputs).

Case studies and discussions

- It is still not clear from the model description, what the evaluated DMUs are? Is it one Shandong Vocational and Technical University of International Studies, where each year is treated as one DMU, or is it a set of universities in a region? From the result description, it is rather one university and each year as a DMU. In this case, what about the discrimination ability (power) of the model?

- Further, you constructed a two-stage model, but you do not present the efficiency results of those two stages. Please, present the results separately.

- In the sections 5.1, 5.2 and 5.3 include some quantitative evidence. You only state, for example, “The university has significantly improved the overall capability of its research team by introducing high-level research talents…” – how? Where is the evidence of the high-research talent? Where can we see it in the data?

- In general, Discussion is missing. The authors only provide several statements without referencing to other studies. What are the implications/suggestions of the analysis? Further, what are the limitations of the analysis?

References

- In the text Wang and Jiang (2019), but in the references Wang, Jiang, and Zheng (2019). Please revise this.

- (Wang, et al.2018) -> (Wang et al., 2018). Please revise all cases of “et al.” in the article.

- “ResearcEXTRA SPACE h Management, 41(04):280-288.”

- Wang (2012) in the text, but W, L.N. (2012) in the References.

- María and Oscar ( 2020) – not included in the references.

- Boon and Andrew (2016) – not included in the references.

- Geraint and Jill ( 2016) – not included in the references.

- Hu et al. (2017) in the text, but Hu, D.X., Wang, Y.W. (2017) in the references.

- Liu et al. (2018) – not included in the references.

- García-Vega, M., Vicente-Chirivella, O. (2020). – not used in the text.

- Johnes, G., Johnes, J. (2016). – not used in the text.

- Lee, B.L., Worthington, A.C. (2016). – not used in the text.

- Liu, W., Gong, S.W. (2018). - not used in the text.

-

Finally, after reading the article, the title does not fully correspond to the body of the article. You do not construct Science and Technology Innovation efficiency evaluation, rather the article deals with an efficiency evaluation at one university. Further, can be such model used at other universities of similar type? Are all indicators available for all universities?

**Do you want your identity to be public for this peer review?** For information about this choice, including consent withdrawal, please see our Privacy Policy

Reviewer #1: **Yes: ** Arif Onan

Reviewer #2: **Yes: ** Martin Flegl

---

## [Author Response · Author response to Decision Letter 1]

17 Dec 2024

Dear Editors and Reviewers:

Thank you for your letter and for the reviewers' comments concerning our manuscript entitled "Construction of Science and Technology Innovation Efficiency Evaluation System for Vocational Undergraduate Colleges" (ID:PONE-D-24-

21161). Those comments are all valuable and very helpful for revising and improving our paper, as well as the important guiding significance to our researches. We have studied comments carefully and have made correction which we hope meet with approval. Revised portion are marked with a different color in the paper.

---

## [Editor Report · Decision Letter 1]

3 Jan 2025

Dear Dr. Zeng,

Thank you for submitting your manuscript to PLOS ONE. After careful consideration, we feel that it has merit but does not fully meet PLOS ONE’s publication criteria as it currently stands. Therefore, we invite you to submit a revised version of the manuscript that addresses the points raised during the review process.

Dear author,

Please see the reviewer 2 suggestions:

Introduction

- Please add a reference related to the 18th CPC National Congress in the first paragraph.

- “…modernization country, and to accelerate…” Split the sentence into two here.

- “In December 2020, the Ministry of Education convened…” – reference is missing. The same in “In May 2019, the Ministry of Education announced the first 15 pilot…”

- Please rearrange chronologically the Introduction. The authors are jumping from September 2020, to December 2020, then to May 2019.

- “Scientific and technological innovation activities of colleges and universities…” – reference(s) is missing in this part. Moreover, you should add a figure to display this process.

- “Although the overall development level of scientific and technological innovation in colleges and universities in China has been significantly improved…” – how? Please, be more specific and provide some statistics, figures, etc., and support your statement with references.

- Further, the same, “there are still problems such as low utilization of innovation resources and unbalanced regional development” – references are missing.

- “Although the overall development level of scientific…” – this sentence has 13 lines! Please, revise this sentence and divide it into several shorter ones to improve the readiness and flow.

- “The main reasons for the inefficiency of innovation are clarified.” – what does it mean? Where?

Research on the influencing factors of scientific and technological innovation efficiency in universities

- This chapter is only one paragraph! Divide the text into several shorter and logical paragraphs.

Research on the evaluation of scientific and technological innovation efficiency in universities

- “…technological innovation efficiency, respectively.DIVIDE THE PARAGRAPH HERE The third category compares…”.

- It is not necessary to explain all forms of the DEA methodology used in the STI evaluation (CCR, BCC, two/three stages, MI and WA models. It is enough saying that the non-parametric approach based on the DEA methodology is used, and continue directly: “In this case, Shen and Gong (2013) used…”

Selection of indicators

- In table 1, please better divide the inputs, intermediates and outputs, use a simple line between each category. Right now one can think, for example, that the intermediates begin on the line next to Intermediate output indicators. The same with the Additional input indicators.

- The authors mention that “…the result transformation stage include part of the initial input resources”. – Which of the initial inputs are later partially used? It is not clear from Table 1.

- The authors refer to Table 4.1 and 4.2, but I do not see these tables in the article. Rather, Table 1, etc. In this case, Table 4.2 refers to Table 2? I am not sure about it.

- Could you please explain a bit how the Y6j Research waste output was treated in the model as it has negative values? Further, I do not understand how you transformed the negative values in Table 2 to positive normalized values in Table 3! Did you treat this variable as an undesirable variable? If yes, how exactly?

- Regarding the normalized values, why 1.2 of 2020 values was selected? Why not to use as a basis 2013 as 1.000 and then correspondingly normalize the rest of the years (such as GDP in prices of XXXX year.

Model building

- “ g additional inputs H (h=1,2,...,g), and g new inputs H (h=1,2,...,g). ...,q), g new inputs Hhj (h=1,2,...,g),” – first, I guess this description is an error. Second, does this description want to say that there are additional new inputs and shared inputs to Stage 2? If yes, use different parameters letter (for example, g additional inputs and r shared inputs).

Case studies and discussions

- It is still not clear from the model description, what the evaluated DMUs are? Is it one Shandong Vocational and Technical University of International Studies, where each year is treated as one DMU, or is it a set of universities in a region? From the result description, it is rather one university and each year as a DMU. In this case, what about the discrimination ability (power) of the model?

- Further, you constructed a two-stage model, but you do not present the efficiency results of those two stages. Please, present the results separately.

- In the sections 5.1, 5.2 and 5.3 include some quantitative evidence. You only state, for example, “The university has significantly improved the overall capability of its research team by introducing high-level research talents…” – how? Where is the evidence of the high-research talent? Where can we see it in the data?

- In general, Discussion is missing. The authors only provide several statements without referencing to other studies. What are the implications/suggestions of the analysis? Further, what are the limitations of the analysis?

References

- In the text Wang and Jiang (2019), but in the references Wang, Jiang, and Zheng (2019). Please revise this.

- (Wang, et al.2018) -> (Wang et al., 2018). Please revise all cases of “et al.” in the article.

- “ResearcEXTRA SPACE h Management, 41(04):280-288.”

- Wang (2012) in the text, but W, L.N. (2012) in the References.

- María and Oscar ( 2020) – not included in the references.

- Boon and Andrew (2016) – not included in the references.

- Geraint and Jill ( 2016) – not included in the references.

- Hu et al. (2017) in the text, but Hu, D.X., Wang, Y.W. (2017) in the references.

- Liu et al. (2018) – not included in the references.

- García-Vega, M., Vicente-Chirivella, O. (2020). – not used in the text.

- Johnes, G., Johnes, J. (2016). – not used in the text.

- Lee, B.L., Worthington, A.C. (2016). – not used in the text.

- Liu, W., Gong, S.W. (2018). - not used in the text.

Finally, after reading the article, the title does not fully correspond to the body of the article. You do not construct Science and Technology Innovation efficiency evaluation, rather the article deals with an efficiency evaluation at one university. Further, can be such model used at other universities of similar type? Are all indicators available for all universities?

We look forward to receiving your revised manuscript.

Kind regards,

Alexandre Morais Nunes, Ph.D.

Academic Editor

PLOS ONE

---

## [Author Response · Author response to Decision Letter 2]

7 Jan 2025

Dear PLOS ONE Editorial Team,

Thank you for your valuable feedback on my manuscript titled "Construction of Science and Technology Innovation Efficiency Evaluation System for Vocational Undergraduate Colleges" (PONE-D-24-21161R2). I have addressed the requested revisions as follows:

1.I have updated the title in the online submission system to ensure it matches the title in the manuscript.

2.I have made the necessary references to Figure 1 and Figure 2 in the text, and these references have been highlighted in red for easy identification.

I have resubmitted the revised manuscript for your review. Please let me know if any further adjustments are required.

Thank you for your continued consideration.

Best regards,

Jun Zeng

---

## [Editor Report · Decision Letter 2]

8 Jan 2025

Evaluation of Scientific and Technological Innovation Efficiency at Shandong Vocational and Technical University of International Studies: A Two-Stage DEA Model Approach

PONE-D-24-21161R2

Dear Dr. Zeng

We’re pleased to inform you that your manuscript has been judged scientifically suitable for publication and will be formally accepted for publication once it meets all outstanding technical requirements.

Kind regards,

Alexandre Morais Nunes, Ph.D.

Academic Editor

PLOS ONE

Additional Editor Comments (optional):

The authors responded to each remark through detailed, point-by-point answers and implemented the necessary revisions in the manuscript.

---

## [Editor Report · Acceptance letter]

PONE-D-24-21161R2

PLOS ONE

Dear Dr. Zeng,

I'm pleased to inform you that your manuscript has been deemed suitable for publication in PLOS ONE. Congratulations! Your manuscript is now being handed over to our production team.

Kind regards,

on behalf of

Professor Alexandre Morais Nunes

Academic Editor

PLOS ONE